# Electrochemical Biosensors for Tracing Cyanotoxins in Food and Environmental Matrices

**DOI:** 10.3390/bios11090315

**Published:** 2021-09-04

**Authors:** Antonella Miglione, Maria Napoletano, Stefano Cinti

**Affiliations:** 1Department of Pharmacy, University Naples Federico II, Via Domenico Montesano 49, 80131 Naples, Italy; Antonella.miglione@unina.it (A.M.); marianapoletano0@gmail.com (M.N.); 2BAT Center–Interuniversity Center for Studies on Bioinspired Agro-Environmental Technology, University Naples Federico II, 80055 Naples, Italy

**Keywords:** electroanalysis, screen printed electrodes, voltammetry, impedance, aptamer, microcystin-LR, anatoxin-a, saxitoxin, cylindrospermopsin

## Abstract

The adoption of electrochemical principles to realize on-field analytical tools for detecting pollutants represents a great possibility for food safety and environmental applications. With respect to the existing transduction mechanisms, i.e., colorimetric, fluorescence, piezoelectric etc., electrochemical mechanisms offer the tremendous advantage of being easily miniaturized, connected with low cost (commercially available) readers and unaffected by the color/turbidity of real matrices. In particular, their versatility represents a powerful approach for detecting traces of emerging pollutants such as cyanotoxins. The combination of electrochemical platforms with nanomaterials, synthetic receptors and microfabrication makes electroanalysis a strong starting point towards decentralized monitoring of toxins in diverse matrices. This review gives an overview of the electrochemical biosensors that have been developed to detect four common cyanotoxins, namely microcystin-LR, anatoxin-a, saxitoxin and cylindrospermopsin. The manuscript provides the readers a quick guide to understand the main electrochemical platforms that have been realized so far, and the presence of a comprehensive table provides a perspective at a glance.

## 1. Introduction

Some strains of cyanobacteria, also known as blue-green algae, can produce toxins (cyanotoxins) that represent huge danger to humans and animals, contaminating drinking water, water used in agricultural irrigation, for recreational purposes and for cultivating or simply supporting the life of aquatic species. Anthropogenic activity and global warming are identified as the main factors involved in the growing presence of harmful algal blooms [1,2,3,4]. According to their toxic effect, cyanotoxins are mainly classified as follows: (i) hepatotoxins (microcystins and nodularins): they are implicated in the inhibition of phosphate proteins 1A and 2A, which cause hyperphosphorylation of cytoskeletal filaments, deformation of hepatocytes, cancer promotion and liver damage, (ii) neurotoxins (anatoxin-a, anatoxin-a(s), saxitoxins and analogs, and β-methylamino-l-alanine): they are low molecular weight alkaloids that block sodium channels by inhibiting nerve conduction, and (iii) cytotoxins (cylindrospermopsin): involved in the inhibition of glutathione, protein synthesis and are responsible for necrotic and genetic damage [5,6]. Although those listed are the most present, dermatoxins (lyngbyatoxin, aplysiatoxin and debromoaplysiatoxin) and irritating toxins (lipopolysaccharid endotoxins) can also be present, and responsible for skin irritation and inflammation of the gastrointestinal tract, respectively. Microcystins are the most widespread cyanotoxins, and can be found worldwide, e.g., in the United States, China, Australia, New Zealand, Germany, Romania, Spain, Sweden, Turkey and Italy. To date, almost 80 variants of microcystins have been identified, each with different polarity, lipophilia and toxicity [7]. Among them, the microcystin-LR, with leucine (L) and arginine (R) as variable amino acids, is the most widespread and most toxic congener. World Health Organization (WHO) guidelines indicate 1 μg/L in drinking water and a tolerable daily consumption (TDI) of 0.04 μg/kg per day for MC-LR [8,9,10].

The main route of exposure is usually accidental ingestion, but cyanotoxins can also be aerosolized under certain conditions, as they have been found in the nasal passages of coastal residents [11,12,13]. Microcystins are produced by Microcystis, a species of cyanobacteria typical of freshwater basins and is generally responsible for toxin-related concerns. Other toxins, such as anatoxin-a and cylindrospermopsins, are produced by Anabaena and a number of other freshwater cyanobacteria species [14,15]. Saxitoxins can also be produced by several species of marine dinoflagellates [16]. These 4–5 groups generally represent the most discussed cyanobacterial toxins. However, ca. 70 congeners of these toxins have been isolated [17], thus confirming the necessity of establishing a depth monitoring in order to assure safety. To ensure the quality of water and food and to preserve human health, several methods of detection and quantification for cyanobacteria have been developed.

Conventionally, methods such as enzyme-linked immunosorbent assay (ELISA) and liquid chromatography–mass spectrometry (LC–MS) are the most used techniques for detecting and quantifying cyanotoxins. For this reason, the United States Environmental Protection Agency (USEPA) has adopted these two techniques, ELISA in “Method 546” and LC–MS in “Method 544”, as official methodologies describing protocols for detection of total microcystins and nodularins in water samples [18,19]. LC–MS is used when high sensitivity is needed, and to differentiate congeners within a toxin group. ELISA is most often employed for total toxin quantification. However, LC–MS is more expensive and involved than ELISA, which is more time efficient and economical by comparison. In addition to these approaches, mouse bioassays, enzymatic tests, electrophoresis, HPLC also offer common strategies even if their major drawbacks are mainly due to their complex experimental setup, specialized personnel required, ethical issues, high cost and long procedures [20,21,22,23,24,25,26,27,28,29,30,31,32,33,34,35]. For these reasons, an alternative to traditional assays is represented by biosensors, which are simple, economical and efficient tools for detecting plethora of pollutants, including natural toxins [36,37,38]. Among the different strategies that have been employed for cyanotoxins detection using biosensors, e.g., colorimetric, electrochemical, fluorescence, plasmonic, the electrochemical ones have appeared as the most suitable for decentralized monitoring due to their unique features such as miniaturization, high compatibility with portable commercial readers (e.g., PalmSens developed a smartphone-powered potentiostat) and being not affected by colored/opaque matrices [39,40,41]. Electrochemical approaches have been highly powered by the adoption of nanomaterials and synthetic recognition probes (e.g., aptamers), which have been able to manufacture highly sensitive and specific platforms for handheld monitoring.

In this review, we would like to highlight some of the recent electrochemical approaches that have been reported in the biosensor’s community with direct application to cyanotoxins determination in environmental and food fields. In particular, the review focuses on three classes of cyanotoxins, namely hepatotoxins, neurotoxins and cytotoxins, e.g., microcystin-LR, anatoxin-a, saxitoxin, cylindrospermopsin (chemical structures are reported in Figure 1). Recent examples are described, and a comprehensive table is reported to provide the readers a quick view of the possible strategies for developing an electrochemical (bio)sensor for cyanotoxin detection.

## 2. Microcystins

In this section, the most common approaches to detect microcystin-LR, anatoxin-a, cylindrospermopsin and saxitoxin, are reported and discussed. However, prior to begin with the discussion of sensing strategies, it should be noted how the principal electrochemical-based methods are based on voltammetric/amperometric, impedimetric and potentiometric architectures. Briefly, voltammetric/amperometric detection produces a response as a consequence of a specific redox reaction occurring at the working electrode, impedimetric measurements mostly quantify the change of the resistance of the charge transfer (of an external electrochemical probe) at the working electrode when some binding (probe-target) occurs, and potentiometric approaches are based on the measurement of the electrical potential of a working electrode when no current is flowing. It should be anticipated that, with regards to the probes that are mainly utilized to recognize cyanotoxins, oligonucleotide aptamers appear as the most utilized, both for voltammetric/amperometric and impedimetric measurements. In particular, these molecules (selected after a combinatorial process, namely SELEX) are able to specifically bind a particular cyanotoxin. The recognition leads to a conformational change which is reflected with a increase/decrease current flow through the working electrode (if a redox mediator is linked to the aptamer, and the conformational change of the aptamer makes the redox mediator closer or farther from the working electrode surface). Instead, when an aptamer-based impedimetric architecture is developed, the cyanotoxin-aptamer adduct produces a detectable change due to the fact that an external mediator can be hindered or enhanced in exchanging electrons with the working electrode surface. An alternative to aptamers, enzymes (i.e., acetylcholinesterases) can be used as the recognition probes. In this case, the presence of the cyanotoxin (inhibitor) reduces the substrate conversion at the enzyme. If the enzymatic by-product is an electroactive molecule, the presence of cyanotoxin will be revealed as a decrease of the recorded current produced by the by-product, with respect to the current recorded in presence of just the enzymatic substrate (in absence of inhibitor). This approach is particularly suggested for amperometric detection.

### 2.1. Microcystin-LR

Among the hepatotoxins, microcystins (MCs) represent a group of monocyclic species that are produced by the flowering of cyanobacteria. Their structure is commonly composed of seven amino acids, five constants and two variables, including the amino acid called 3-amino-9-methoxy-2,6,8-trimethyl-10-phenyldeca-4,6-dienoic acid (Adda), the only one associated with their toxicity.

Of all MCs, microcystin-LR (MC-LR) containing leucine (L) and arginine (R), is the most widespread and toxic variant; in fact, we find it mainly in fresh water but also in brackish and marine water.

Eissa and co-workers [42] developed a voltammetric aptasensor to determine the presence of MC-LR in tap water and fish samples. The architecture consisted of the combination of specific oligonucleotide-based aptamer and graphene onto an electrochemical platform. In this case, the interaction between the two components was not covalent, but the synthetic probe was just drop casted onto the carbonaceous substrate. The sensing architecture involved a physical adsorption of the specific aptamer to graphene hexagonal cells through π−π stacking interactions. The author observed an increase of the voltammetric peak in presence of the target, i.e., MC-LR, as shown in Figure 2A. The presence of MC-LR induces a conformational change of the aptamer, leading to a dissociation some aptamer molecules from the surface of the electrode, then decreasing the total negative charge on the graphene surface and improving the accessibility to external redox couple (ferro/ferricyanide) towards the electrode area, producing a high current. On the contrary, the absence of MC-LR did not allow a satisfactory electron transfer due to the presence of negatively charged aptamers, which electrostatically repelled the redox couple (negatively charged). The sensing architecture appeared highly specific (in presence of okadaic acid, MC-LA and MC-YR) and MC-LR was detected down to 1.9 pM under the optimal experimental conditions.

In Lebogang’s work [43], another approach was used: in particular, to determine the presence of MC-LR in freshwater samples, an amperometric flow ELISA system, called VersAFlo, was produced, as shown in Figure 2B. The system was fully automated, and the recognition was based on the presence of monoclonal antibodies linked to sepharose beads and packed into a micro-immunocolumn. Inside the column, free microcystin and microcystin-horseradish peroxidase conjugate (important for the following production of electrochemical signal) were sequentially captured by the immobilized antibodies. The subsequent addition of the enzymatic substrate, namely 2,20-azinobis (3-ethylbenzothiazoline-sulfonic acid) (ABTS), leads to an enzymatic oxidation producing ABTS°+ that is electrochemically reduced at the working electrode. Due to the described competitive mechanism, the intensity of the reduction peak decreased in presence of high levels of MC-LR (without peroxidase conjugation). Authors highlighted how the use of VersAFlo system was capable of reducing the analysis time (20 min for each cycle of analysis) and providing information in real-time. The electrochemistry was not affected by the presence of possible interferences and a satisfactory detection limit of 0.01 mg/L has been obtained.

Another interesting approach was also reported by Lin and colleagues [44], that immobilized a MC-LR aptamer onto a gold electrode by exploiting Au-S chemistry. Briefly, an impedimetric detection was adopted, and the impedance decreased as a consequence of the binding between MC-LR and aptamer onto the sensing electrode. The presence of MC-LR led to an adduct formation which resulted in the impedance decreasing, with respect to the absence of MC-LR. The decrease rate had a linear relationship with the logarithm of the MC-LR concentration with a detection limit of 18 pM. The sensor has been applied to detect MC-LR in three kinds of real water samples (lake, river, tap) with good selectivity and stability, and recoveries comprised 91 and 113%. The use of magnetic particles (MPs) has been also used to improve detection of MC-LR. As reported by Reverte’s work [45], MPs have been used as supports for the immobilization of biorecognition molecules for the detection of MC-LR. The MPs were used as a support for the immobilization of biorecognition molecules for the detection of MC-LR. The G protein-coated MPs are conjugated to a monoclonal antibody (mAb) against MC-LR and a direct enzymatically competitive immunoparticle test (ELIPA) was then performed. The ELIPA strategy has provided limits of detection of 3.9 μg/L of MC-LR. The approach has been applied to the analysis of a cyanobacterial culture and a natural bloom, and MC equivalent contents have been compared with those obtained by conventional assays and liquid chromatography-tandem mass spectrometry (LC–MS/MS). Results have demonstrated the viability of the use of MPs as biomolecule immobilization supports in biotechnological tools for MCs monitoring. In Table 1, additional methods for microcystin-LR detection are highlighted [46,47,48,49,50,51,52].

### 2.2. Anatoxin-a

Anatoxin-a is a highly dangerous neurotoxin produced by some freshwater cyanobacteria during flowering. It represents a cholinergic agonist that binds to acetylcholine receptors in the central and peripheral nervous system and neuromuscular junctions, causing continuous stimulation with blockage of electrical transmission. Exposure to anatoxin-a commonly occurs by ingesting or drinking contaminated food and water, which induces clinical signs such as salivation, tearing, incontinence, defecation, convulsions and cardiac arrhythmia. Death occurs from muscle paralysis and asphyxiation; toxic effects are observed in humans, animals, birds, and fish.

In the work carried out by Elshafey [53], a DNA aptamer was selected and characterized for developing a label-free impedimetric aptasensor for neurotoxin anatoxin-a. The SELEX procedure started by incubation of the DNA library of 10^15^ random 60 nucleotide sequences with ATX-beads, and the chosen aptamer (ATX-8) was chosen due to the highest affinity towards the anatoxin-a, and a Kd of ca. 80 nM was calculated. As reported in Figure 3A, the presence of ATX led to a decrease in the impedance due to a better electron transfer between the external electrochemical probes and the gold electrode surface. Authors estimated the limit of detection equal to 0.5 nM at a signal-to noise of 3, and it was lower than that of the commercial ATX receptor-binding assay as well as the guideline value of 1 μg/L for water safety [54]. ATX-8 shows several advantages than other natural or artificial receptors, such as determination of diversified targets, chemical stability, flexibility and selectivity, in fact the stability of the aptasensor (examined by storing the sensor buffer solution at 4 °C for 15 days) was demonstrated by a 89% retention of its original impedance signal.

Another possibility to detect anatoxin-a is represented by the adoption of inhibition-based enzymatic biosensors, very well used for the detection of pesticides [56]. In particular, due to the inhibitory effect of anatoxin-a towards cholinesterase enzyme, different strategies have been developed focusing on the design of portable electrochemical biosensors by revealing the inhibition towards acetylcholinesterase activity (AChE) [57]. Devic and co-workers [55] evaluated the presence of anatoxin-a in freshwater samples by exploiting the AChE inhibition. Since the inhibition is irreversible, the reaction occurs until all anatoxin-a has reacted with the enzyme. A prerequisite of a biological test is the selective activity, but this does not occur with the inhibition of AChE because artificial toxins such as organophosphates and carbamates, used as insecticides, can also inhibit the activity of the enzyme, Figure 3B. For this reason, the production of a selective test involved the use of mutant enzymes sensitive to anatoxin-a and resistant to most insecticides. In this case, the enzyme was engineered through a physical entrapment onto the screen printed working electrode, using a photo-cross-linkable poly vinyl alcohol. All measurements were performed after the enzyme-engineered electrode was immersed in an aqueous solution for 10 min. The two sensitive enzymes (mutants) allowed the detection limit down to 0.5 nM, and the electrochemical platform was satisfactorily applied to diverse cyanobacterial bloom samples from Spain, Greece, France, Scotland and Denmark. Another example regarding the application of electrochemical inhibition-based electrochemical biosensor for anatoxin-a detection has been reported by Villatte et al. [58]. In this case, the graphite working electrode included an electrochemical mediator within the formulation, namely 7,7,8,8-tetracyanoquino-dimethane, in order to promote the charge transfer in presence of the electroactive enzymatic by-product, thiocholine. In this work, the electric eel enzyme was used, as it turns out to be the most sensitive to anatoxin-a. The developed system displayed a detection limit of 1 ppb, and authors demonstrated how the reactivation (through the use of oxime) was not possible after anatoxin-a inhibition (0% reactivation) while it was possible after inhibition in presence of organophosphorous pesticide, i.e., paraoxon, leading to a ca. 70% reactivation. This result could be exploited to differentiate the presence of cyanotoxin and pesticide in water samples.

### 2.3. Saxitoxin

Saxitoxin or STX is the best-known paralytic shellfish toxin (PST). The toxin is naturally produced by certain species of marine dinoflagellates (*Alexandrium* sp., *Gymnodi*-*nium* sp., *Pyrodinium* sp.) but also by cyanobacteria (*Anabaena* sp., some *Aphanizomenon* spp., *Cylindrospermopsis* sp., *Lyngbya* sp., *Planktothrix* sp.). Ingestion of saxitoxin, usually by consumption of shellfish contaminated by toxic algal blooms, is responsible for the life-threatening paralytic shellfish poisoning (PSP). PSP toxins including STX and its analogs, e.g., neosaxitoxin and gonyautoxins, are potent neurotoxins that might block mammalian voltage-gated sodium channels resulting in death, ca. 2000 cases/year. Toxin analysis is currently performed with mouse bioassay, in vitro functional and cell assays, ELISA, and a number of analytical techniques involving liquid chromatography coupled to mass spectrometry [59,60,61]. Due to the importance of contamination, rapid and preferably on-site detection is necessary. Biosensors might offer a suitable alternative for STX detection and various approaches have been proposed in literature.

In the work carried out by Bratakou [62], a miniaturized saxitoxin potentiometric sensor based on graphene nanosheet is described in which the Anti-STX, the natural saxitoxin receptor, is immobilized on embedded lipid films. An adequate selectivity for STX detection over a wide range of toxin concentrations, in presence of several ions interferents (magnesium, calcium, chloride, ammonium etc.) was achieved. The response time ranges from 5 to 20 min and the system exhibits a stable response within the dynamic range 10^−9^–10^−6^ (pH = 7.0) with a detection limit of 1 nM. The method was successfully applied for the detection of saxitoxin in simulated and real lake waters, and shellfish samples. The lake water samples, spiked with a 50 nM and 1 mM range of saxitoxin, give recoveries of between 90 and 102%, while the measurements carried out onto mussels, mollusks and oysters gave recoveries comprised between 103 and 107% (Figure 4A).

Instead, in Hou’s work [63] a label-free electrochemical aptasensor was developed for selective detection of STX. In this case, the aptamer was covalently bonded to a multi-walled carbon nanotube film (MWCNT) which coated an octadecantiol monolayer which was deposited on a gold electrode (Figure 4B). Methylene blue, used as an electrochemical mediator (MB), was electrostatically anchored on the carboxylated MWCNTs. A strong differential pulse voltammetric signal is produced when the target is missing (STX), while when STX binds to its aptamer, it triggers a conformational change of the aptamer resulting in the formation of a barrier that prevents the MB from reaching the electrode surface.

The peak oxidation current of MB decreases linearly with increasing STX concentrations in the concentration range of 0.9 and 30 nM. The detection limit is 0.38 nM. The assay was applied to the determination of STX in mussel samples and was found to be acceptably accurate (RSD = 8%). Furthermore, the aptasensor also demonstrated good storage stability when stored at 4 °C under dry conditions, maintaining 90% of its initial response signal after 2 weeks. In addition, a magnetic electrochemical immunosensor for ultra-sensitive detection of STX in seawater and seafood was developed in Jin’s work [64]. This immunosensor consists of STX-specific antibody-functionalized magnetic beads (MBs) for STX recognition, palladium-doped graphitic carbon nitride (g-C3N4-PdNPs) mimetic peroxidase to catalyze hydrogen peroxide-mediated oxidation of 3.3’, 5,5’-tetramethylbenzidine (TMB) to generate signal. The proposed immunosensor was used to detect traces of STX in seawater and crustacean samples, reaching a detection limit of 1.2 pg/mL and a recovery of 93–107% with RSD (*n* = 5) <5%. In addition to these approaches, an impedimetric-based architecture for detecting saxitoxin has been highlighted in Table 1 [65].

### 2.4. Cylindrospermopsin

Cylindrospermopsin (CYN) is one of the cyanotoxins of greatest concern due to its potential toxicity and spread to various environments, including drinking water. CYN has potential interference with human and animal metabolic pathways, which affect the functions of organs including liver, kidneys, lungs, etc. CYN is involved in inhibiting protein synthesis and detaching ribosomes from the membrane of the endoplasmic reticulum. It also interacts with soluble proteins, which are associated with protein translations, and cytochrome p450 is believed to be responsible for the rapid toxicity of CYN. In Elshafey’s work [66], to determine the presence of cylinderspermopsin in freshwater samples, an unlabeled and highly sensitive aptasensor was used based on the use of aptamers as specific receptors. DNA aptamers were selected from different libraries using the SELEX in vitro screening approach and they showed high affinity for CYN (Kd in the nanomolar range). Authors highlighted how the immobilization of the aptamer onto a surface produced a change in the dissociation constant compared to the free aptamer in fluorescence studies. The aptamer, conjugated with a gold-based electrochemical surface, was used to manufacture an impedimetric aptasensor, through the formation of a self-assembled monolayer from a disulfide-derivatized aptamer on a gold electrode. The detection mechanism is based on the change of impedance in presence of an external electrochemical mediator, when the CYN-aptamer adduct was formed. A detection limit of 100 pM and a wide linear range up to 80 nM have been obtained, demonstrating high selectivity in presence of other cyanotoxins like microcystin-LR and Anatoxin-a.

Valério and colleagues reported about preliminary findings on the determination of cylinderspermopsin in freshwater samples, using an electrochemical DNA biosensor based on electrodes modified with polytiramine (PTy) [67]. A coding sequence of cylinderspermopsin (PKSM4 5’-phosphate probe) has been covalently linked to polythyramine using N-hydroxysuccinimide (NHS) and 1-etyl-3-(3-dimethylaminopropyl)-carbodiimide hydrochloride (EDC) as coupling agents. Methylene blue (MB) was used as electroactive indicator and the quantification of the target was carried out by monitoring the increase of the voltammetric peak in presence of MB. In order to evaluate the correct hybridization, fluorescence experiments (using PicoGreen as the DNA dye) have been carried out, and the recognition was confirmed. However, further informations have been reported regarding the affinity of the immobilized DNA probe in a subsequent publication reported by the same group, even if a direct application towards the cyanotoxin is still missing. In particular, as reported in Figure 5A, the same group of authors demonstrated that voltametric characterization of the modified electrodes in standard solution in presence of methylene blue as the redox probe, has allowed to confirm the presence of a cylindrospermopsin-producing strain probe and following hybridization with a complementary sequence [68].

A recent study, carried out by Zhao and colleagues [69] describes the development of a label-free impedimetric aptasensor for detecting CYN. In this case, the recognition probe is represented by an amino-substituted aptamer covalently bonded to a thionine–graphene (TH–G) nanocomposite through the use of glutaraldehyde as a crosslinking agent. The presence of CYN causes a change in the conformation of the aptamer, which results in a significant decrease in the resistance to electron transfer, which thus increases the impedance (Figure 5B). Under optimum conditions, CYN was quantified in a wide range between 0.39 and 78 ng/mL, with a good correlation of 0.9968, and a low detection limit of 0.117 ng/mL. Regarding the presence of 1 nM interfering toxins such as okaidaic acid and mycrocistin-LR, the response was negligible in comparison with that produced by the presence of cylindrospermopsin. The developed procedure allowed to obtain a batch-to-batch reproducibility equal to 1.2%, demonstrating the good suitability of the proposed platform for further investigations.

In the following Table 1, all the major characteristics for the electrochemical sensing of the most relevant cyanotoxins are reported. Readers have provided a quick guide to develop novel electrochemical architectures.

Regarding the main experimental performances and analytical figures of merits, it should be highlighted how all the experimental procedures are characterized by mean times of ca. 10–20 min to perform the electrochemical detection. In particular, the adoption of aptamers as the recognition probes, require some minutes to allow the binding occurs. In addition, some approaches (in particular those based on impedance) are characterized by the necessity of washing steps (perhaps to decrease unspecific binding on the electrodes’ surface). Regarding the sensitivity, a satisfactory method to lower the detection limit is represented by the use of pre-concentration (e.g., magnetic beads), but this reduces the easiness of the approach, and in addition, it represents an ulterior task for the end-user. However, depending on the different mechanisms, there is no evidence of a favorite approach over the others, in fact it should be noted that all the studies have been carried out in spiked-samples, thus further development should be evaluated in really polluted samples.

## 3. Discussion and Future Outlooks

The increasing stringency of water legislation towards the surveillance of environmental sites has highlighted the role of portable analytical devices. In particular, decentralized electrochemical biosensors offer a great possibility towards the on-field application for natural toxins detection such as cyanotoxins. The interest in developing electrochemical biosensors for algal toxin detection during the last decade has increased, as specifically highlighted by comprehensive reviews that have been reported in the recent 4–5 years [70,71,72].

However, it should be noted how the research on these targets, from a biosensor point of view, is still niche. To date, the use of biosensors are not completely mature to replace the use of traditional and laboratory-bound approaches such as LC–MS based ones, but of course biosensors can represent a valuable complementary toolkit for first screenings. The development of biosensors for cyanotoxins (and natural toxins in general) is dependent on three main challenges: recognition element, sensitivity and experimental settings. The adoption of aptamers represents an obvious step in advance with respect to their natural counterparts, i.e., antibodies. Even if aptamers provide valuable advantages such as stability, possibility to be labelled with electrochemical mediators, avoid the ethical issues with animals’ use, low cost and high affinity, it should be considered that SELEX processes often produce valuable probes for “in-solution” assays that are difficult replicated when engineered onto working electrodes. However, it should be considered that when natural toxins are considered as the target species, a variety of toxins’ congeners could occur while detecting them in real samples. The presence of different (small) functional groups in large molecules, e.g., microcystins, might be translated into a lower binding activity for the designed aptamer. The use of a single aptamer could not be sufficient to assess the real risk in complex matrices, but an array of these should be developed in future. To this aim, emergent chemometrics (multivariate analysis), artificial intelligence and machine learning approaches would represent a step forward in the development of multiplexed platforms [73,74,75]. The use of multivariate statistics is capable to preprocessing data, reduce noise and extract hidden correlation among multiple factors, that are often missed with univariate approaches. Another interesting approach, even if not largely reported, is the use of inhibition-based biosensors: even if the costs associated to this approach is lower in comparison with the aptamers, however, it should be considered that cholinesterase enzymes are inhibited in presence of plenty of pollutants, e.g., pesticides, insecticides, metals, and other toxins, and this could affect the selectivity of the platform. In particular, when the biosensors are applied toward agri-food matrices, the presence of the interfering species could represent a major issue. A good strategy, as reported for anatoxin-a detection, is represented by the use of mutated enzymes. In this case the cost for producing these recognition elements should be considered. Another possibility to overcome the challenge related to selectivity, is the use of microfluidic module and smart materials, i.e., nanomotors. As reported in literature, pesticides can be hydrolyzed with the adoption of pre-column or alkaline treatment, and to avoid the presence of inhibiting agents when detecting cyanotoxins, a paper-based microfluidic device could associated as the pre-cleaning step for interferents removal [76,77].

The second major challenge is represented by the sensitivity of the electrochemical platforms. As reported in the text, gold and carbon-based nanomaterials are the most utilized ones. However, no one of the authors reported the use of gold nanoparticles instead of gold-ink/gold rods for making the biosensors. The use of nanomaterials is able to improve both the sensitivity of the final platforms and to reduce the cost of the final devices (an aspect that should be considered prior to realized commercial devices). The combination with novel nanomaterials and nanocomposites should be improved. Regarding the use of impedimetric approaches, for instance, the same approaches are displayed by the authors. A possibility would also be to use different and size-controlled electrochemical mediators, perhaps to improve the electron transfer at the electrode surface. As for other fields of application, i.e., cancer diagnostics, biosensors may reach improved sensitivity if combined with innovative signal transduction nanosystems.

The third major challenge is represented by the long experimental procedures that are needed to perform the quantification of cyanotoxins. Authors reported the time for measurements at the final stage, but a non-specialized reader is not usually informed on how long and delicate are washing steps. In particular, those based on impedance and those based on enzymatic inhibition require multiple washing steps and addition of buffers. End-users require easy-to-use approaches on the basis of the most common biosensors, i.e., glucometer and pregnancy tests. All these steps should be reduced, and the adoption of porous paper-based materials could represent the next generation of electrochemical sensing for detecting these species. In fact, the use of paper-based materials, in particular chromatographic paper, offers the possibility to store all the reagents into ad hoc strips. Paper has the capability to store dry reagents and to filter gross impurities thanks to its characteristic porosity [78,79]. Of course, paper-based substrates do not represent the panacea for electrochemical biosensing, but they can represent an ulterior possibility to make the devices closer to citizens.

## 4. Conclusions

In our opinion, the development of electrochemical biosensors for this class of toxins has still room to grow, and major improvements can be achieved with a combination of rising technologies such as paper-based substrates, chemometrics/artificial intelligence approaches, multi-recognition elements and use of smart-nanomaterials that can improve both the sensitivity and the multiplex ability of future platforms. In the near future, breakthroughs in electrochemical biosensors will certainly contribute to the growth of the emerging field of natural toxins tracing.

## Figures and Tables

**Figure 1 biosensors-11-00315-f001:**
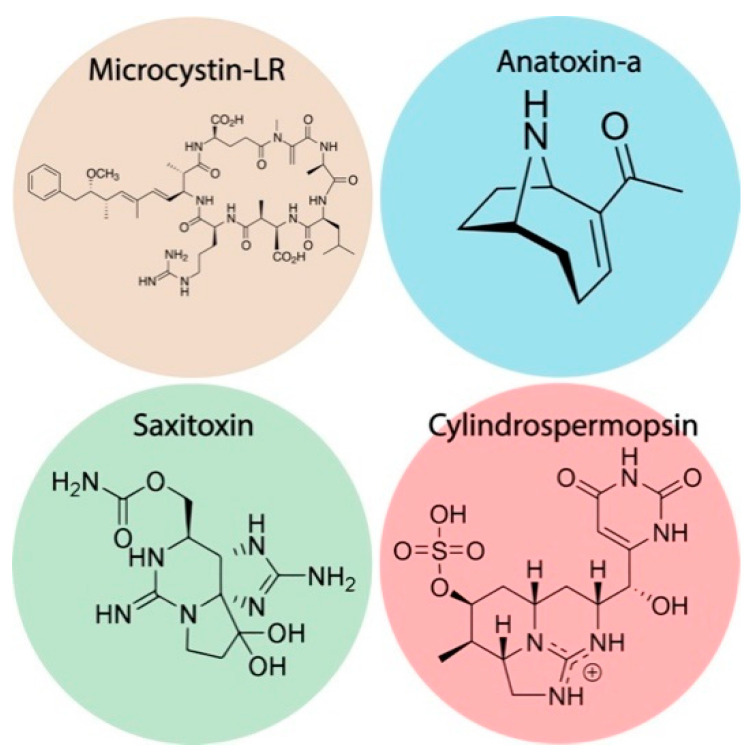
Chemical structures of major relevant cyanotoxins in environmental and food samples.

**Figure 2 biosensors-11-00315-f002:**
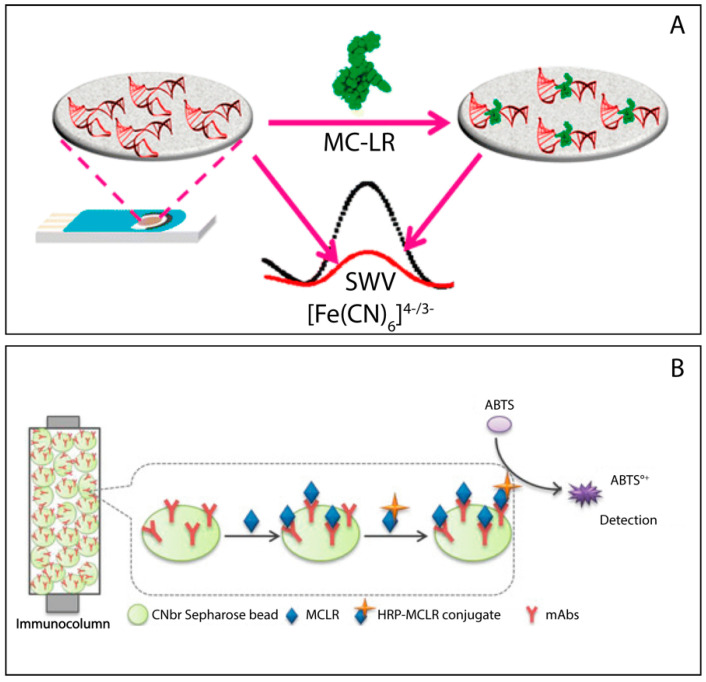
(**A**) Schematic of MC-LR Detection Based on SWV on Aptamer-Functionalized GSPE. Reprinted with permission from Ref. [42]. Copyright 2014, American Chemical Society; (**B**) Schematic demonstration of the extraction taking place in the immunocolumn. Reprinted from ref. [43].

**Figure 3 biosensors-11-00315-f003:**
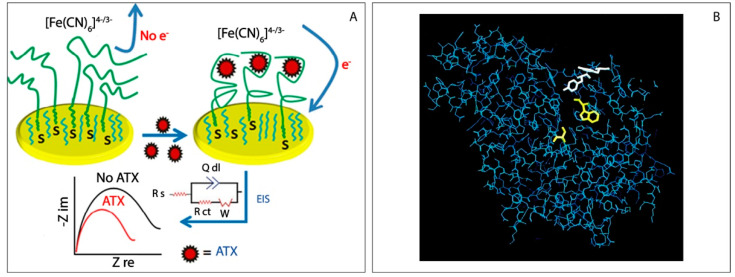
(**A**) Fabrication of the label-free impedimetric aptasensor of anatoxin-a. Without toxin, the negatively charged redox probe is repelled from the surface and its redox reaction is hindered. Upon toxin recognition to the aptasensor, the aptamer induced switching into compact structure, implying more accessible of the electrochemical marker to the surface and the resistance to electron transfer is decreased. Equivalent circuit used to fit the frequency scans along with an impedance spectra. Reprinted with permission from Ref. [53]. Copyright 2015, Elsevier; (**B**) FIG. 2. Drosophila AChE, it shows the positions of mutations that provided the best sensitivities to anatoxin-a(s) (Tyr71 and Tyr73) in white at the entrance of the active site gorge. Amino acids important for catalysis (Ser238 and Trp83) at the bottom of the active site are shown in yellow. Reprinted from ref. [55].

**Figure 4 biosensors-11-00315-f004:**
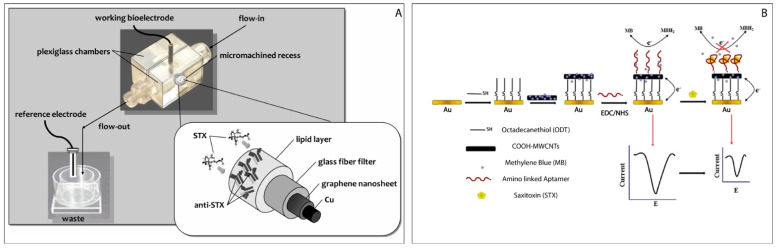
(**A**) Schematic representation of the potentiometric sensor based on graphene sheets for saxitoxin detection. Reprinted with permission from Ref. [62]. Copyright 2016, Wiley; (**B**) The schematic diagram of amperometric aptasensor for saxitoxin using a gold electrode modified with carbon nanotubes on a self-assembled monolayer, and Methylene Blue as an electrochemical indicator probe. Reprinted with permission from Ref. [63]. Copyright 2016, Springer.

**Figure 5 biosensors-11-00315-f005:**
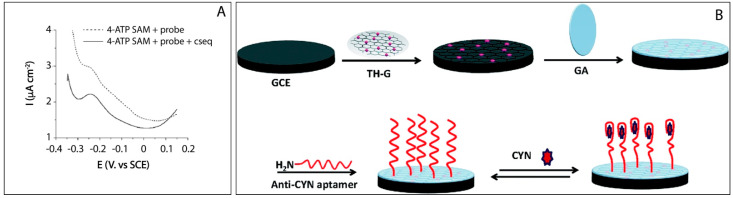
(**A**) Square-wave voltammograms of 4-ATP SAM modified electrodes after probe covalent attachment and hybridization with a complementary sequence, after MB accumulation. Reprinted with permission from Ref. [68]. Copyright 2008, Wiley; (**B**) Schematic representation of the label-free impedimetric aptasensor for cylindrospermopsin detection. Reprinted with permission from Ref. [69]. Copyright 2015, Royal Society of Chemistry.

**Table 1 biosensors-11-00315-t001:** Brief schematization of the electrochemical biosensors reported for microcystin-LR, anatoxin-a, saxitoxin and cylindrospermopsin detection in various matrices.

Cyanotoxin	Electrochemical Method	Sensing Platform	Electrode Materials/Nanomaterials	LOD	Time of Measure	Real Matrix	Ref
Microcystin-LR	Voltammetry	Aptamer assembled on a modified electrode with graphene	Carbon nanomaterial graphene	0.0019 nM	10 min	Drinking water	[42]
Microcystin-LR	Amperometry	Immobilized monoclonal antibodies on Sepharose beads	Sepharose beads on a screen printed electrode	10 nM 1000 nM	20 min	Fresh water	[43]
Microcystin-LR	Impedance	Modified electrode (Au-S)	Self-assembled monolayer	0.018 nM	10 min	Fresh water	[44]
Microcystin-LR	Amperometry	Monoclonal antibodies conjugated to protein G-coated MPs	Carbon screen printed electrode	3.9 μg/L	10 min	Cyanobacterial culture and a natural bloom	[45]
Microcystin-LR	Photoelectrochem.	Photoelectrode of graphene doped with nitrogen	BiOBr nanoflakes/N-doped graphene p–n heterojunction electrode	3.0 × 10^−5^ nM	15 min	Fish	[46]
Microcystin-LR	Voltammetry	Drugged aptamer with electro-synthesized silver nanoparticles	AgNPs with cobalt(II) salicylaldiiminemetallodendrimer on a glassy carbon electrode	0.04 μg/L	10 min	Fresh water	[47]
Microcystin-LR	Elisa	Nitrogen-doped carbon nanotubes assembled on gold nanoparticles	AuNPs; Nitrogen-doped carbon nanotubes (Au/CNx-MWNTs)	0.004 μg/L	10 min	Fresh water	[48]
Microcystin-LR	Elisa	Polyclonal antibodies (produced in sheep)	SPE with a membrane containing an immobilized isoproturon-ovalbumin conjugate	0.06 μg/L	10 min	Fresh water	[49]
Microcystin-LR	Impedance	3D graphene-based electrochemical impedance spectroscopy biosensor using antibodies.	3D graphene foam (GF) sheets as working electrode	0.05 mg/L	30 min	Drinking water supply	[50]
Microcystin-LR	Voltammetry	Electrochemical detection of MC-LR based on infinity-shaped DNA structure using double aptamer and terminal deoxynucleotidyl transferase	Screen printed gold electrode	15 pM	90 min	Serum and tap water samples	[51]
Microcystin-LR	Amperometry (direct method) Voltammetry (indirect method)	Dual-mode aptasensor based on MoS2-PtPd (direct method) and ZIF-8-Thi-Au (indirect method)	GCE modified with MoS2-PtPd-NPs or (ZIF)-8-thionine (Thi)-Au	0.006 ng/mL 0.045 ng/mL	60 min	Water environment	[52]
Anatoxin-a	Impedance	Modified disulfide aptamer assembled on a gold electrode	Self-assembled monolayer	0.05 nM	10 min	Drinking water	[53]
Anatoxin-a	Amperometry	Inhibition enzymes (inhibition of acetylcholinesterase activity)	Graphite working electrode	0.5 nmol/L	10 min	Fresh water	[55]
Anatoxin-a	Amperometry	Inhibition enzymes (inhibition of acetylcholinesterase activity)	7,7,8,8,-tetracyanoquinodimethane (TCNQ)-graphite working electrode	1 μg/L	10 min	Drinking water	[58]
Saxitoxin	Potentiometry	Anti-STX incorporated lipid films on graphene nanosheets	Graphene nanosheets. Anti-STX, immobilized on a stabilized lipid films	1 nM	5 min	Lake water samples, fresh shellfish samples.	[62]
Saxitoxin	Amperometry	Gold electrode modified with carbon nanotubes on a self-assembled monolayer	Monolaer of octadecanethiol deposited on a gold electrode, coated with MWCNTs	0.38 nM	30 min	Mussel samples	[63]
Saxitoxin	Amperometry	STX-specific antibody-functionalized magnetic beads (MBs). Palladium-doped graphitic carbon nitride (g-C3N4-PdNPs) peroxidase mimetic	Palladium-doped graphitic carbon nitride nanosheets on a magnetic gold electrode	1.2 pg/mL	75 min	Seawater and shellfish samples	[64]
Saxitoxin	Impedance	Label-free impedimetric aptasensor	Self-assembled monolayer on Au electrode	0.3 μg/L	60 min	Aqueous solution	[65]
Cylindrospermopsin	Impedance	Modified disulfide aptamer assembled on a gold electrode	Self-assembled monolayer from a disulfide-derivatized aptamer on a gold electrode	0.1 nM	10 min	Fresh water	[66]
Cylindrospermopsin	Voltammetry	Electrodes modified with polytiramine (PTy)	Pt disk	25 pg/mL	10 min	Fresh water	[67]
Cylindrospermopsin	Impedance	Covalent immobilization of the aptamer of CYN on the thionine–graphene (TH–G) nanocomposite	Glassy carbon electrode	0.117 ng/mL	120 min	Lake water	[69]

## Data Availability

Not applicable.

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
