# Peer review of "Electrochemical Biosensors for Tracing Cyanotoxins in Food and Environmental Matrices"

_biosensors, 2021, doi:10.3390/bios11090315_

Round 1

Reviewer 1 Report

This review focuses on electrochemical approaches for cyanotoxins biosensing in environmental and food samples. The relevance of the detection of four common cyanotoxins together with recent examples of biosensors for these toxins were adequately explained. The authors discuss the main challenges and perspectives regarding this topic, which is of interest to researchers in the field of biosensors. In my opinion, the following suggestions should be addressed:

- Line 58: It is not clear the idea the authors aim at communicating in that sentence

- A title for section 2 may be added

- It would be useful to include the electrode materials/nanomaterials used for all sensors in Table 1.

- The analytical characteristics of the biosensor in ref. 52 collected in the table should be revised, as they does not seem to correspond to that reference.

- The authors may cite recent comprehensive reviews on electrochemical biosensors for cyanotoxins/toxins, such as ACS Sens. 2019 May 24; 4(5): 1151–1173, DOI: 10.1021/acssensors.9b00376 and ACS Sens. 2018, 3, 7, 1233–1245, DOI: 10.1021/acssensors.8b00359

Author Response

We are grateful to all the comments and criticisms raised by the reviewers because they gave us the opportunity in improving the quality of the manuscript, by including more information and discussion, which can be very useful to a large audience. We reported the “reasons” expressed by reviewers and the answer (in red) to their questions.

Reviewer 1

This review focuses on electrochemical approaches for cyanotoxins biosensing in environmental and food samples. The relevance of the detection of four common cyanotoxins together with recent examples of biosensors for these toxins were adequately explained. The authors discuss the main challenges and perspectives regarding this topic, which is of interest to researchers in the field of biosensors. In my opinion, the following suggestions should be addressed:

- Line 58: It is not clear the idea the authors aim at communicating in that sentence

Accorgind to suggestion, line 58 has been re-written as follows:

“However, ca. 70 congeners of these toxins have been isolated [17], thus confirming the necessity of establishing a depth monitoring in order to assure safety.”

- A title for section 2 may be added

Title has been added, namely “2. Microcystins”

- It would be useful to include the electrode materials/nanomaterials used for all sensors in Table 1.

Table 1 has been modified accordingly in the revised form of the manuscript.

- The analytical characteristics of the biosensor in ref. 52 collected in the table should be revised, as they does not seem to correspond to that reference.

The reference has been removed from the table. Its new ref. number is 57, and it is a review that is still cited in the text.

- The authors may cite recent comprehensive reviews on electrochemical biosensors for cyanotoxins/toxins, such as ACS Sens. 2019 May 24; 4(5): 1151–1173, DOI: 10.1021/acssensors.9b00376 and ACS Sens. 2018, 3, 7, 1233–1245, DOI: 10.1021/acssensors.8b00359

Accordingly, we cited three recent reviews including the two suggested by the reviewer, in the revised manuscript as follows (Section 3):

“The increasing stringency of water legislation towards the surveillance of envi-ronmental sites, has highlighted the role of portable analytical devices. In particular, de-centralized electrochemical biosensors offer a great possibility towards the on-field application for natural toxins detection like cyanotoxins. The interest for developing electrochemical biosensors for algal toxin detection during the last decade has increased, as specifically highlighted by comprehensive reviews that have been reported in the recent 4-5 years [70-72].”

Reviewer 2 Report

In this work (manuscript biosensors-1349145), the authors discuss the use of electrochemical biosensors in cyanotoxins quantification in food and environmental samples. Selected applications of electrochemical biosensors in the detection of some cyanotoxins, namely hepatotoxins, neurotoxins and cytotoxins, are presented. The work is well organized and the peculiarities of each class of toxins are described. The topic pertains to the journal’s aim and scope and it is of interest in the environmental and food fields.

The manuscript can be considered for publication after minor revision:

  1. The functioning principle of electrochemical biosensors based on impedance, voltammetric, amperometric, and potentiometric detection modes should be briefly described at the beginning of section 2. The title of section 2 is missing.
  2. A brief description of the sensing principles of the aptasensors and inhibition-based enzymatic biosensors should be included in section 2.
  3. The main advantages of various electrochemical detection methods displayed in Table 1 should be briefly described in the text. The main figures of merit of the analytical performance presented in Table 1 like limit of detection and time of measure should be also discussed in the text.
  4. Relevant papers published in the last five years should be included in Table 1.

Author Response

We are grateful to all the comments and criticisms raised by the reviewers because they gave us the opportunity in improving the quality of the manuscript, by including more information and discussion, which can be very useful to a large audience. We reported the “reasons” expressed by reviewers and the answer (in red) to their questions.

Reviewer 2

In this work (manuscript biosensors-1349145), the authors discuss the use of electrochemical biosensors in cyanotoxins quantification in food and environmental samples. Selected applications of electrochemical biosensors in the detection of some cyanotoxins, namely hepatotoxins, neurotoxins and cytotoxins, are presented. The work is well organized and the peculiarities of each class of toxins are described. The topic pertains to the journal’s aim and scope and it is of interest in the environmental and food fields.

The manuscript can be considered for publication after minor revision:

  1. The functioning principle of electrochemical biosensors based on impedance, voltammetric, amperometric, and potentiometric detection modes should be briefly described at the beginning of section 2. The title of section 2 is missing.

Accordingly, we added title of Section 2 and added the principle of electrochemical biosensors as follows:

“In this section the most common approaches to detect Microcystin-LR, anatoxin-a, cylindrospermopsin and saxitoxin, are reported and discussed. Hovewer, prior to begin with the discussion of sensing strategies, it should be noted how the principal electrochemical-based methods are based on voltammetric/amperometric, impedimetric and potentiometric architectures. Briefly, voltammetric/amperometric detection produces a response as a consequence of a specific redox reaction occurring at the working electrode, impedimetric measurements mostly quantify the change of the resistance of the charge tranfer (of an external electrochemical probe) at the working electrode when some binding (probe-target) occurs, and potentiometric approaches are based on the measurement of the electrical potential of a working electrode when no current is flowing.

  1. A brief description of the sensing principles of the aptasensors and inhibition-based enzymatic biosensors should be included in section 2.
  2. Accordingly, section 2 has been improved with brief descriptions, as follows:
  3. “It should be anticipated that, with regards to the probes that are mainly utilized to recognize cyanotoxins, oligonucleotide aptamers appear as the most utilized, both for voltammetric/amperometric and impedimetric measurements. In particular, these molecules (selected after a combinatorial process, namely SELEX) are able to specifically bind a particular cyanotoxin. The recognition leads to a conformational change which is reflected with a increase/decrease current flow through the working electrode (if a redox mediator is linked to the aptamer, and the conformational change of the aptamer makes the redox mediator closer or farer from the working electrode surface). Instead, when an aptamer-based impedimetric architecture is developed, the cyanotoxin-aptamer adduct produces a detectable change due to the fact that an external mediators can be hindered or enhanced in exchanging electrons with the working electrode surface. In alternative to aptamers, enzymes (i.e. acetylcholinesterases) can be used as the recognition probes. In this case, the presence of the cyanotoxin (inhibitor) reduces the substrate conversion at the enzyme. If the enzymatic by-product is an electroactive molecule, the presence of cyanotoxin will be revealed as a decrease of the recorded current produced by the by-product, with respect to the current recorded in presence of just the enzymatic susbtsrate (in absence of inhibitor). This approach is particularly suggested for amperometric detection.”

  1. The main advantages of various electrochemical detection methods displayed in Table 1 should be briefly described in the text. The main figures of merit of the analytical performance presented in Table 1 like limit of detection and time of measure should be also discussed in the text.

As suggested, the principal performance has been added in the revised text of the manuscript, as follows:

Regarding the main experimental performances and analytical figures of merits, it should be highlighted how all the experimental procedures are characterized by mean times of ca. 10-20 minutes to perform the electrochemical detection. In particular, the adoption of aptamers as the recognition probes, require some minutes to allow the binding occurs. In addition, some approaches (in particular those based on impedance) are characterized by the necessity of washing steps (perhaps to decrease unspecific binding on the electrodes’ surface). Regarding to the sensitivity, a satisfactory method to lower the detection limit is represented by the use of pre-concentration (e.g. magnetic beads), but this reduce the easiness of the approach, and in addition it represents an ulterior task for the end-user. However, depending on the different mechanisms, there is no evidence of a favorite approach over the others, in fact it should be noted that all the studies have been carried out in spiked-samples, thus further development should be evaluated in really polluted samples.

  1. Relevant papers published in the last five years should be included in Table 1.

Table 1 has been revised accordingly, and inserted in the revised version of the manuscript.
